# Topography-induced large-scale antiparallel collective migration in vascular endothelium

Claire Leclech [1], David Gonzalez-Rodriguez [2], Aurélien Villedieu[3], Thévy Lok[1], Anne-Marie Déplanche[4] & Abdul I. Barakat [1✉]

Collective migration of vascular endothelial cells is central for embryonic development, angiogenesis, and wound closure. Although physical confinement of cell assemblies has been shown to elicit specific patterns of collective movement in various cell types, endothelial migration in vivo often occurs without confinement. Here we show that unconfined endothelial cell monolayers on microgroove substrates that mimic the anisotropic organization of the extracellular matrix exhibit a specific type of collective movement that takes the form of a periodic pattern of antiparallel cell streams. We further establish that the development of these streams requires intact cell-cell junctions and that stream sizes are particularly sensitive to groove depth. Finally, we show that modeling the endothelial cell sheet as an active fluid with the microgrooves acting as constraints on cell orientation predicts the occurrence of the periodic antiparallel cell streams as well as their lengths and widths. We posit that in unconfined cell assemblies, physical factors that constrain or bias cellular orientation such as anisotropic extracellular matrix cues or directed flow-derived shear forces dictate the pattern of collective cell movement.

[1] LadHyX, CNRS, Ecole Polytechnique, Institut Polytechnique de Paris, Palaiseau, France. [2] Université de Lorraine, LCP-A2MC, F-57000 Metz, France. [3] Institut Curie, Université PSL, Sorbonne Université, CNRS UMR 3215, Inserm U934, Genetics and Developmental Biology, 75005 Paris, France. [4] Nantes Université, École Centrale Nantes, CNRS, LS2N, UMR 6004, F-44000 Nantes, France. ✉email: abdul.barakat@polytechnique.edu

Collective cellular movement is central for many physiological and pathological processes including morphogenesis, angiogenesis, wound healing, and cancer invasion[1]. A prominent manifestation of collective cell behavior is the emergence of complex migration patterns whose structure and organization depend on many factors, including the physical constraints to which the cells are subjected. For instance, mesoscale geometric confinement of cellular monolayers has been shown to organize otherwise chaotic cell movement into directed bidirectional motion, swirling, or even global rotation, with the specific pattern dependent on cell type and the size and shape of the confinement area[2–7].

Although geometric confinement is representative of some physiologically relevant scenarios, collective migration in vivo often occurs without such confinement. A prominent example is vascular endothelial cell wound healing following device- or disease-induced injury. In this case, the migrating cell sheets remain free of global confinement, but the cells' basal surfaces are nonetheless subjected to anisotropic biophysical cues due to the nano- to microscale topography of the thin fibers of the underlying extracellular matrix[8,9]. How these ubiquitous biophysical cues regulate collective cell migration patterns remains unknown.

Here we address this question by studying the collective motion of vascular endothelial cells on anisotropic microgroove substrates that constitute idealized mimics of extracellular matrix topography[10]. We show that unconfined endothelial monolayers on microgrooves exhibit a specific type of collective migration that takes the form of periodic antiparallel cell streams whose characteristic dimensions are considerably larger than either the groove or cell size, suggesting that these topographic constraints induce long-range effects. We also show that the emergence of the cell stream pattern requires intact cell-cell contacts and that the sizes of the streams are determined by the groove dimensions, most notably groove depth. Additionally, we demonstrate that the streams develop in a monolayer that exhibits minimal spatial heterogeneities in cell activity, no sustained cell polarization, and highly dynamic cell-cell rearrangements. Modeling the endothelial cell sheet as an active fluid with the microgrooves acting as constraints on cell orientation accurately predicts the occurrence of the periodic antiparallel cell streams, as well as their lengths and widths. Finally, we demonstrate that other types of biophysical cues that have an impact on cell orientation, such as directional fluid flow, give rise to the same cell stream pattern. These results indicate that external physical factors that exert subcellular-scale physical constraints on individual cells within an unconfined monolayer can drive large-scale collective movement patterns.

## Results

**Endothelial cells on microgroove substrates migrate with a periodic pattern of antiparallel cell streams.** The movement of monolayers of human umbilical vein endothelial cells (HUVECs) was monitored on polydimethylsiloxane (PDMS) microstructured substrates composed of arrays of parallel grooves of width (w), spacing (s), and depth (d) of 5 μm, uniformly coated with fibronectin (Fig. 1a and Supplementary Fig. 8a). Flat PDMS substrates served as controls. In line with previous reports[11–13], endothelial cells on the microgroove substrates aligned and elongated in the groove direction (labeled as the x-axis) (Fig. 1a) and exhibited long and oriented focal adhesions (FAs) positioned along the ridges (Supplementary Fig. 1)[13,14]. At confluence, HUVEC nuclei were stained with the live-cell dye Hoechst and recorded for 24 h (Supplementary Movie 1). While nuclei on the flat control surfaces moved in all directions with swirling-like trajectories, cell movement on the microgroove substrates was constrained along the groove direction (Fig. 1b and Supplementary Movie 1), leading to highly directional migration (Fig. 1c). A striking observation was the presence of a specific pattern of movement characterized by periodic streams of cells moving in opposite directions (i.e. antiparallel) along the groove axis with a typical stream width of 100–150 μm (Fig. 1b), considerably greater than either the groove (5 μm) or cell (~30 μm) size. This finding suggests that subcellular-scale physical constraints have wide-ranging ramifications on the organization and migration dynamics at the scale of the entire monolayer. Interestingly, the positions of the antiparallel cell corridors remained relatively constant in time (see the cumulative trajectories, Fig. 1b or the averaged x-velocity fields, Supplementary Fig. 2 and Movie 2).

The extent of instantaneous cell-cell coordination was assessed by comparing the orientation angle of the displacement of each cell at a given time to those of its neighbors within a distance of 100 μm (similar to Hayer et al.[15]). On flat surfaces, only local domains of aligned and coordinated movement were observed (Fig. 1b, d), similar to other studies on endothelial cells[7,15,16] and reminiscent of nematic domains described for other cellular systems[3,17,18]. In contrast, on grooved substrates, a high degree of cellular coordination was observed within the cell streams, leading to overall higher and longer-range instantaneous coordination compared to flat surfaces (Fig. 1e). Cell movement within confluent monolayers involves cell-cell rearrangements, creating cell-scale shearing. The high levels of cell-cell coordination in the cores of the streams led to small x-direction cellular shear rates there, whereas high shear rates were generated locally at the stream borders due to the antiparallel movement (Fig. 1f). Consequently, the average monolayer x-direction shear rate and the average shear gradient in the y-direction were significantly higher on microgroove substrates than on flat substrates (Fig. 1g). Shear forces on the endothelial apical surface due to blood flow are known to trigger myriad signaling events that exquisitely modulate vascular structure and function;[19–21] however, the shear rates in that case are on the order of 100-1000 s$^{-1}$, several orders of magnitude larger than the cell-scale shear rates reported here. It remains unclear if the shear rates and shear gradients induced by the cell stream pattern have significant functional ramifications. If so, potentially interesting targets might be the remodeling of cell-cell junctions and the potential impact on endothelial permeability and barrier function.

**The spatial features of the cell streams are modulated primarily by groove depth.** We then investigated the influence of groove dimensions on the cell streams. Reducing groove depth from 5 to 1 μm while maintaining groove width and spacing constant (5 μm) led to a wider range of cell orientation angles relative to the groove axis (Fig. 2a) and wider but shorter antiparallel cell streams with more poorly defined borders (Fig. 2a, b). To quantify these observations, we computed the time- and x-averaged cell velocity as a function of the cross-stream coordinate y ($\bar{v}_x(y)$; Fig. 2c). Because of the antiparallel nature of the streams, $\bar{v}_x(y)$ exhibits directional oscillations around zero. Reducing the groove depth led to less pronounced oscillations, indicating a weakening of the stream structure (Fig. 2c). As expected, the oscillations were essentially absent on the flat surfaces, reflecting the absence of cell streams. The median stream width, computed as the y-distance between zero crossings in the oscillations, increased from ~100 μm to ~150 μm as groove depth decreased from 5 to 1 μm (Fig. 2d). To quantify the length of the streams and their x-direction persistence, we considered the length over which the Pearson correlation coefficient between increasingly distant columns of the x-velocity field decreased from its original value of 1 to 0.2. Stream length on 1 μm-deep grooves was ~300 μm and increased to

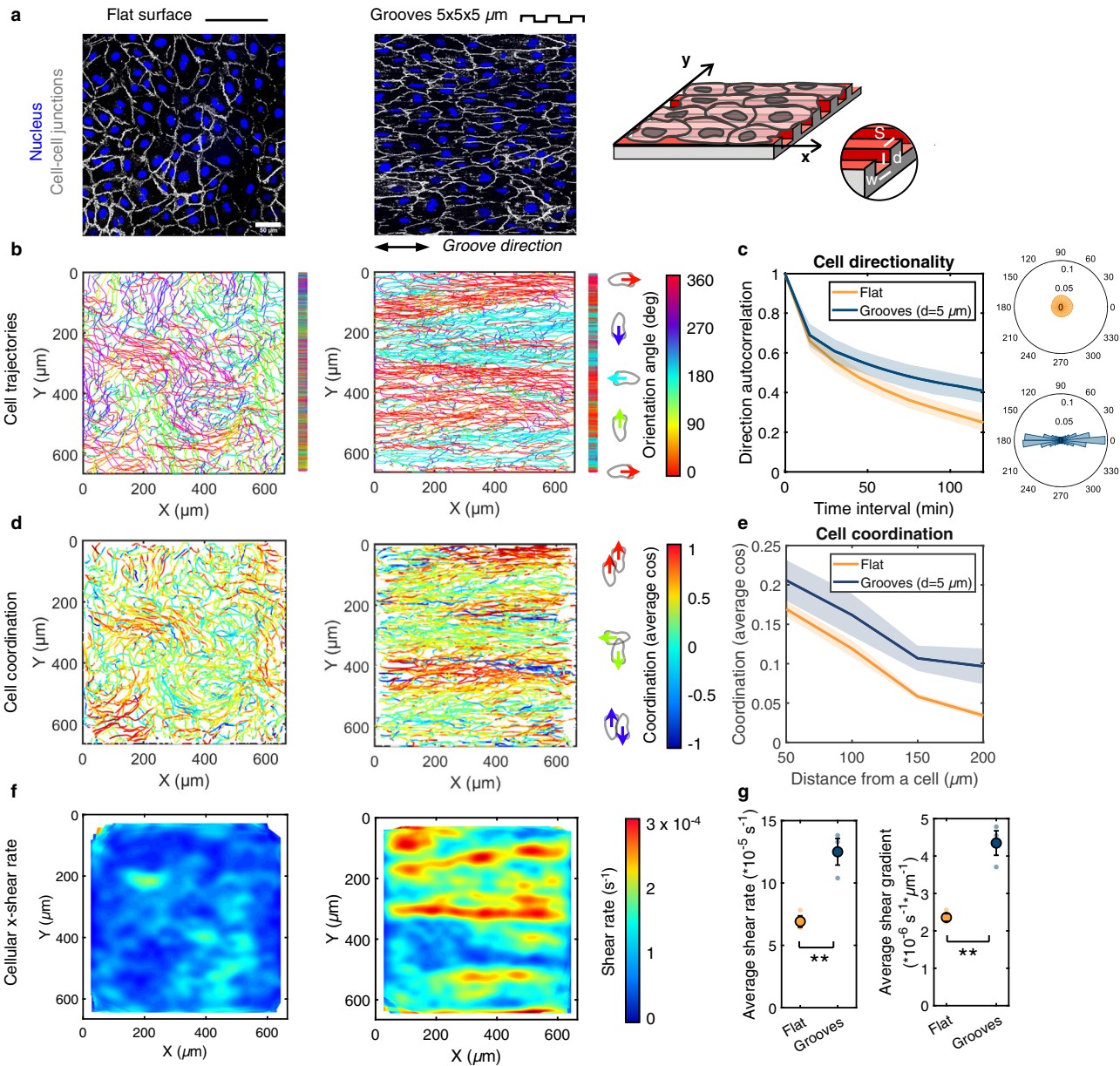

**Fig. 1 Endothelial cells on grooved substrates migrate with a periodic pattern of antiparallel cell streams. a** Monolayers of endothelial cells (HUVECs) on a control flat substrate or substrate with grooves of width, spacing, and depth of 5 μm, immunostained for VE-cadherin (cell-cell junctions, white) and DAPI (nucleus, blue). Scale bar, 50 μm and double-headed black arrow indicates groove direction. The grooved substrate is shown schematically on the right. **b** Accumulated cell trajectories after 24 h of migration color-coded for the orientation of each displacement vector. Vertical bars to the right of the trajectory maps represent the projected trajectories on the y-axis for better visualization of the cell streams. **c** Direction autocorrelation analysis: the cosine of the angular difference between increasingly more distant vectors within the trajectory is computed and averaged for all cells (left). Polar histograms on the right show the distribution of displacement orientation angles. **d** Cell trajectories color-coded for cell-cell coordination, defined as the cosine of the angular difference between a cell and its neighbors within a 100 μm radius. Values of 1 and -1 indicate parallel and antiparallel displacements, respectively. **e** Evolution of the coordination parameter with increasingly distant neighbors. **f** Heatmaps of cellular x-direction shear rate, extracted from the x-velocity field. **g** Average x-direction shear rate and shear gradient in the y-direction. Dots represent individual experiments. Unpaired t-test (** $p < 0.001$). For all graphs, $n = 3$ independent experiments and error bars represent SEM (standard error of the mean). Source data are provided as a Source Data file.

~450 μm on 5 μm-deep grooves (Fig. 2e). Interestingly, for a given groove depth, the stream length and width were markedly less sensitive to groove width and spacing than to groove depth (Supplementary Fig. 3). These findings suggest that the progressively higher constraint on the orientation of cell movement imposed by increased groove depth correlates with the transition from local regions of coordinated movement seen on flat surfaces to the longer-range interactions that lead to regular and persistent cell streams oriented in the direction of the grooves.

**The emergence of cell streams is dependent on cell-cell contacts**. Emergence of collective patterns of movement often requires cell-cell communication through cell-cell junctions that mature with increased cell density[5,22]. To elucidate the potential role of cell-cell contacts in the establishment of the antiparallel streams, nuclear movement on 5 μm-deep grooves was recorded for different cell densities ranging from isolated (individual) endothelial cells to confluent monolayers. Individual cells traced highly linear trajectories in the groove direction with no bias

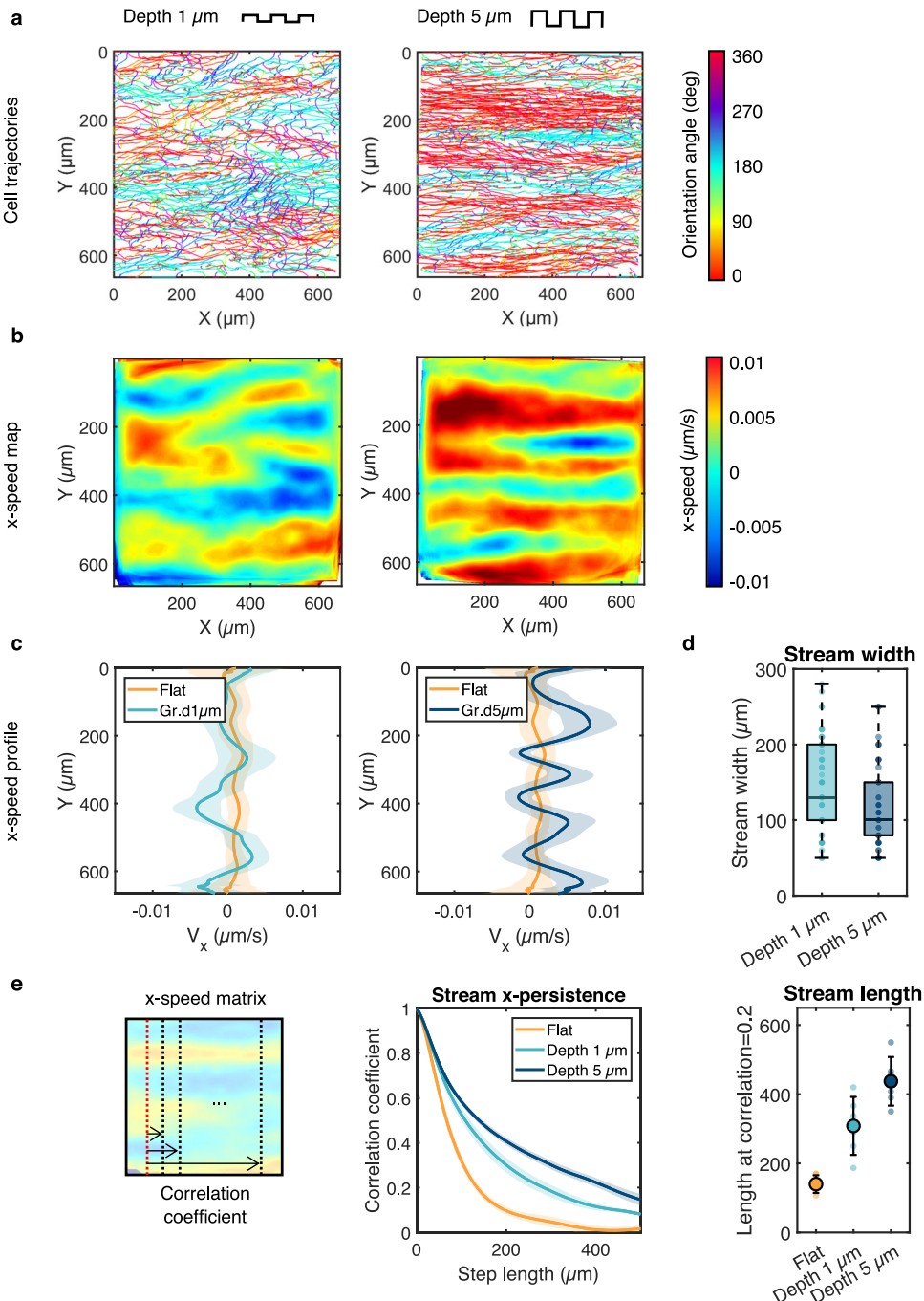

**Fig. 2 The spatial features of the streams are modulated by the groove depth. a** Cell trajectories after 24 h of migration on grooves of width and spacing of 5 μm and depth of either 1 μm (left) or 5 μm (right) color-coded for orientation of displacement vectors. **b** Heatmaps of the x- speed $V_x$, averaged in time. **c** Corresponding $V_x$ profiles along the y-axis averaged over time. Error bars represent standard deviations. **d** The width of the streams was quantified as the average width of the peaks (average distance between zero crossings) in the $V_x$ profiles. Nonparametric *t*-test (**, $p < 0.001$) $n = 61$ streams (depth 1 μm), $n = 76$ streams (depth 5 μm) from 6 independent experiments. **e** Left: the stream persistence along the x-axis was estimated by calculating the Pearson correlation coefficient between increasingly distant columns of the $V_x$ matrix. Error bars represent SEM. Right: the length of the streams was subsequently estimated as the distance corresponding to a correlation coefficient of 0.2. $n = 6$ independent experiments, one-way ANOVA, Tukey's post-test (**, $p < 0.001$). Error bars represent standard deviations. Source data are provided as a Source Data file.

between left-right or right-left movement (Supplementary Fig. 4). The cell streams initiated in low-density monolayers and developed over time as the cell density increased and cell junctions matured, with faster establishment of clearly defined and stable streams for high-density monolayers (Supplementary Fig. 4a, b and Supplementary Movie 3). These observations point towards an important role for cell-cell contacts in the establishment of the stream pattern. To further test this notion, we treated the cells with the calcium chelator ethylenediaminetetraacetic acid (EDTA; 2 and 5 mM) to disrupt cell-cell junctions during the recording time on 5 μm-deep grooves. Immunofluorescence staining confirmed that EDTA treatment strongly diminished the junctional recruitment of VE-cadherin (Supplementary Fig. 4c). Importantly, EDTA treatment disrupted the stream pattern in a dose-

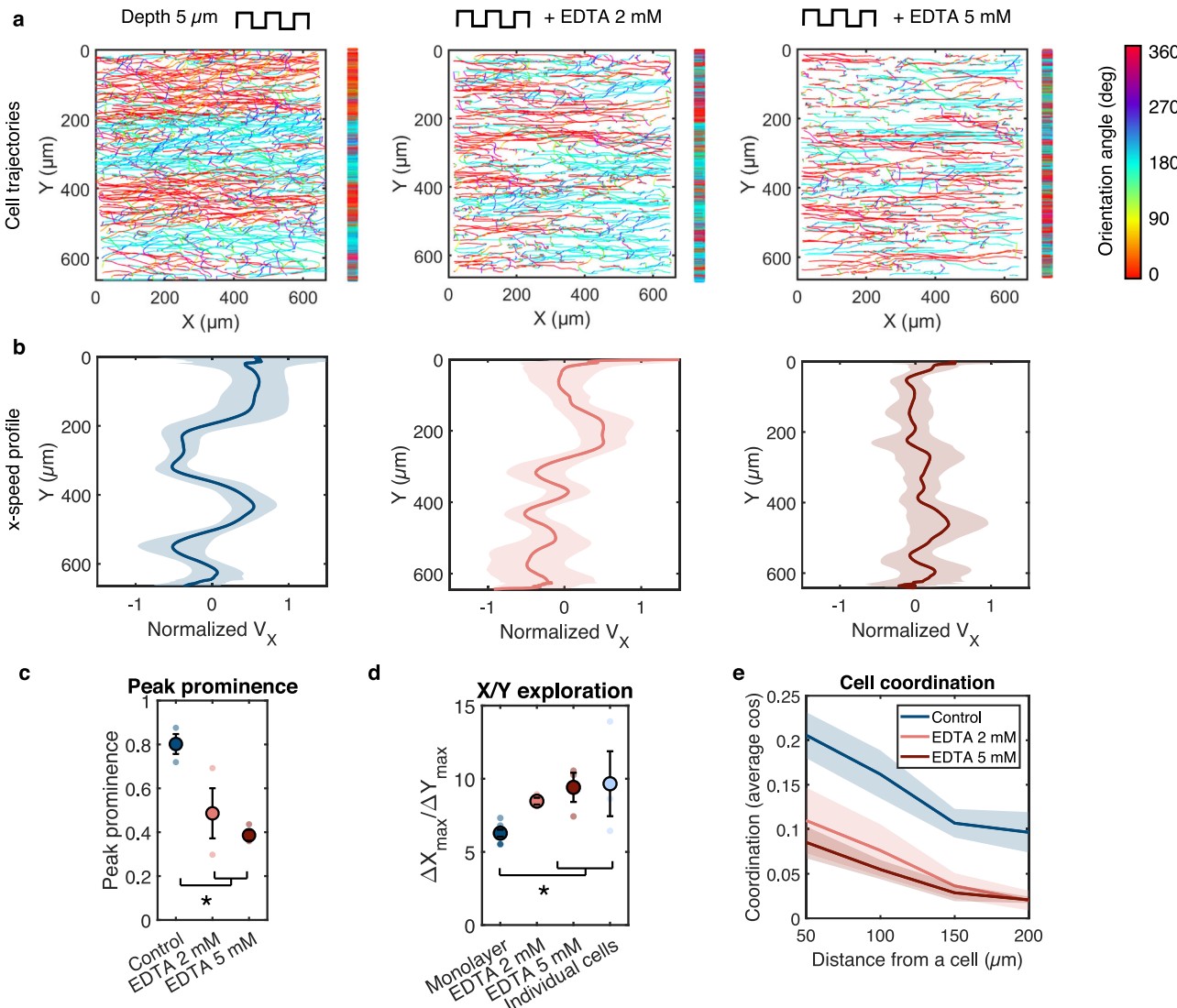

**Fig. 3 The emergence of cell streams is dependent on cell-cell contacts. a** Cell trajectories after 24 h of migration on grooves of width, spacing, and depth of 5 μm with and without EDTA treatment. Trajectories are color-coded for orientation of displacement vectors. Vertical bars represent the projected trajectories on the y-axis. **b** Corresponding $V_x$ velocity profiles (normalized by the mean speed) along the y-axis averaged over time. Error bars represent standard deviations. **c** Peak prominence of $V_x$ profiles (defined as the relative peak height compared to the surrounding baseline). $n = 3$ independent experiments, one-way ANOVA, Tukey's post-test (*, $p < 0.01$). **d** Ratio between x- and y-displacement (calculated as $|X\text{max} - X\text{min}|/|Y\text{max} - Y\text{min}|$ for each trajectory). The value for individual cells is shown for comparison. Monolayer $n = 8$ independent experiments, EDTA and individual cells $n = 3$ independent experiments. One-way ANOVA, Fishers's post-test (*, $p < 0.01$). **e** Evolution of the cell coordination parameter with increasingly distant neighbors. Control $n = 6$ independent experiments, EDTA $n = 3$ independent experiments. For all graphs, error bars represent SEM. Source data are provided as a Source Data file.

dependent manner (Fig. 3a). Consistent with this observation, the amplitudes of the $\bar{v}_x$ oscillations were diminished (Fig. 3b, c). As confirmed by the increased ratio of x- to y-displacement relative to the untreated monolayer (Fig. 3d), EDTA-treated cells exhibited more straight trajectories along the groove axis (Fig. 3a), a behavior similar to that of individual cells on grooves (cf: Supplementary Fig. 4). EDTA treatment significantly decreased cell-cell coordination (Fig. 3e), resulting in cells moving independently of neighboring cells and thus exhibiting "individual-like" behavior even though they remained in a monolayer. These results show that the loss of cell-cell contacts and coordination in endothelial monolayers on microgroove substrates leads to disruption of the cell streams, indicating that this migration pattern is a form of organized collective motion that requires cell-cell communication.

**The emergence of cell streams is not associated with large heterogeneities in cell activity.** Long range, organized collective motions, and in particular directed movements, often involve heterogeneities in cellular activity. A prominent example is the emergence of leader cells that actively drive collective migration[23,24]. To explore if such a scenario is involved in the emergence of the streams, we first examined possible heterogeneities in the force generation machinery, namely actomyosin and focal adhesions, among cells in the monolayer[25–28]. Immunostaining for actin and phospho myosin light chain (pMLC) in monolayers on 5 μm-deep grooves revealed a broadly homogeneous organization of F-actin, with dense stress fibers along the grooves and throughout the cell (Fig. 4a). Actin and pMLC levels, as quantified by the ratio of normalized fluorescence intensity between individual cells and their neighbors, were fairly constant

**Fig. 4 The emergence of cell streams is not associated with large heterogeneities in cell activity. a** Immunostaining of a HUVEC monolayer for actin (top) and phospho myosin II light chain (pMLC; bottom), with DAPI in cyan, on a flat surface or 5 μm-deep microgrooves. Scale bar 50 μm. Quantification shows the ratio between the normalized integrated intensity of fluorescence in one cell and the mean intensity of its neighbors. $N = 10$ cells, unpaired t-test. **b** Cell polarization on fixed monolayers assessed by immunostaining for TGN46 on a flat surface, 1-μm or 5-μm deep microgrooves, at the edge of a wound perpendicular to 5 μm-deep grooves, or in HUVECs subjected to steady flow (shear stress of 2 Pa) for 24 h on a flat surface. Polar plots show the distribution of orientations of the nucleus-Golgi vectors. Scale bar 50 μm. **c** Dynamics of cell-cell rearrangements quantified by analyzing the neighbors present within 100 μm of each cell at each time point. Each black line represents the time during which the same neighbor is present around the considered cell. Quantification shows the mean time spent with the same neighbor and the neighbor exchange rate (gain or loss of neighbor) for cells on 5 μm-deep grooves within a stream, at the border between two streams, or on a flat surface. $N = 20$–30 cells, one-way ANOVA, Tukey's post-test. For all graphs, error bars represent standard deviations. Source data are provided as a Source Data file.

across the monolayer (Fig. 4a). A similar result was obtained for FA area, a surrogate for traction forces exerted by a cell on its substrate[27,28] (Supplementary Fig. 5). Significant heterogeneities in cell elongation and cell-cell junction morphology were also not observed (Supplementary Fig. 6). Taken together, these results point toward the absence of major heterogeneities in contractility and traction forces among cells within the monolayer.

Another prominent feature of leading cells in collective migration is cell polarization[23,29,30]. Analysis of the nucleus-Golgi polarization vectors revealed that unlike cells under flow or in a wound-healing assay which showed clear polarization consistent with previous reports[31–33], endothelial cells on microgrooves exhibited no preferential polarization (Fig. 4b). Thus, sustained cell polarization is not necessary for the cell

alignment, directional migration, and antiparallel streams observed on microgrooves.

From a dynamic perspective, the presence of groups of leading cells in the streams would be expected to be associated with long-term cohesiveness among the cells within the monolayer and limited cellular rearrangement, at least between leaders and their followers. Tracking the evolution of the neighbors around cells during the 24 h recording period revealed highly dynamic cell-cell rearrangements (relative to the characteristic time associated with cell migration speed of ≈30 μm/h) with a relatively short mean time spent with the same neighbor (≈4 h) and a high rate of neighbor exchange (≈3 per h) (Fig. 4c). This was equally the case for both cells within streams and at stream borders. As a further demonstration of the limited cohesiveness of the monolayer on microgrooves, we imaged the movement of endothelial cells plated on a substrate with a flat domain bordering the microgrooves and observed no propagation of the streams as the cells migrated from the grooves onto the flat surface as well as immediate loss of cell elongation and alignment (Supplementary Fig. 7a, b). Interestingly, an intracellular variation in actin organization was visible in cells positioned partly on grooves and partly on the flat substrate, suggesting a high level of endothelial cellular and intracellular plasticity (Supplementary Fig. 7c).

The various results presented above suggest the absence of a stable leader-follower system within endothelial monolayers and indicate that the stream pattern observed on microgrooves is unlikely to be driven by large heterogeneities in cell activity. Rather, our observations point toward a high level of cellular homogeneity in terms of polarization, contractility, and traction forces within endothelial monolayers, with all cells actively participating in the antiparallel cell streams. Of course, leader-follower-type interactions among cells cannot be entirely excluded but were such interactions to occur, they would do so on spatial and temporal scales that are much smaller than those characterizing the streams.

**Active fluid modeling predicts the emergence of antiparallel cell streams.** To gain insight into the physical basis for the emergence of endothelial cell streams, we developed an active fluid model inspired by that proposed by Duclos et al.[4] who had described the emergence of global directional motion in monolayers confined within adhesive stripes. The model details are provided in the Supplementary Note. A key distinctive feature of our model is that it includes an energetic constraint on cell directionality in order to model the effect of substrate grooves. We model the cell sheet as an active fluid with cells treated as nematic active particles characterized by their orientation angle $\theta$ with respect to the groove direction (the $x$-axis). Treating the cells as nematic rather than polar particles is consistent with previous studies[4], and it is based on the following two experimental observations: (1) the endothelial cells used here do not exhibit any marked front-to-back structural asymmetry, associated with a low intrinsic polarization (see Fig. 4b), and (2) cells can spontaneously reverse their direction of motion without turning the cell body over time scales smaller than the time scale of stream development (Supplementary Movie 4). The active fluid theory includes constitutive relations that link the stress to the cell orientation angle, a continuity equation, and a force balance equation. Moreover, the system seeks to minimize its total effective free energy:

$$\mathscr{F} = \int \left( \frac{K_1}{2}(\nabla \cdot \boldsymbol{p})^2 + \frac{K_3}{2}(\nabla \times \boldsymbol{p})^2 + \frac{\alpha}{2}p_y^2 \right), \qquad (1)$$

where $\boldsymbol{p} = (\cos\theta, \sin\theta)$ is the unit director field, and $K_1$ and $K_3$

are the splay and bend Frank constants, respectively. The two first terms are the classical terms representing energetic costs of variations in the orientation field, and the third term is an additional contribution arising from the energetic cost of any cell misalignment with the grooves, characterized by the parameter labeled $\alpha$ (with units of energy per unit surface). By using the Einstein relation (see the Supplementary Note), this coefficient can be estimated as $\alpha \approx \xi_x D_t / \sigma_\theta^2$, where $\xi_x$ is the substrate friction coefficient, $D_t$ the translational diffusion, and $\sigma_\theta$ the standard deviation of cell orientation angles, which is determined experimentally. The parameter $\alpha$ can thus be determined for any groove geometry and provides a useful, condensed description of the strength of cell alignment imposed by different types of mechanical cues.

Actomyosin activity is accounted for through a contractile force dipole acting on each cell $\zeta\Delta\mu$, where $\Delta\mu$ is the free energy produced by nutrient consumption and $\zeta$ is the reactive coefficient as defined in active gel theory[34]. A nonlinear stability analysis reveals that the hydrodynamic profile is formed by alternating streams (Fig. 5a), a prediction that resembles the experimental velocity profile and that is also consistent with a previous numerical simulation predicting the emergence of alternating lanes in active nematic systems[35]. The predicted width of fully-developed streams $W$, is of the order of

$$W^2 \approx \frac{\pi^2 (1+\nu)\gamma(-\zeta\Delta\mu)}{2\alpha\xi_x} , \qquad (2)$$

where η and γ are the shear and rotational viscosities, respectively, arising from both cell mechanical properties and cell-cell interactions. Typical order-of-magnitude parameter values (see Supplementary Table 2) lead to $W$ on the order of 100 μm, an estimate consistent with the experimental observations (cf: Fig. 2d). We note that Eq. (2) yields the dependence $W \sim \alpha^{-1/2} \sim \sigma_\theta$. Moreover, the model predicts a maximum stream velocity that scales as $V_x \sim \alpha \sim \sigma_\theta^{-2}$ (see Supplementary Note). These predicted dependencies are in excellent agreement with the experimental observations (Fig. 5b).

Rotational diffusion promotes cell misalignment with its neighbors, which leads to a persistence length of cell orientation that may correspond to the observed length of the streams. Two consecutive cells along a stream correspond to a step in a random walk. The persistence length of a stream $L$ can be predicted as the length along the $x$-axis for which the typical offset of the random walk along the $y$-axis is equal to the cell size, which leads to

$$L \approx \frac{a}{\sigma_\theta^2} , \qquad (3)$$

where $a \approx 90\mu m$ is the cell size (typical value on the microgroove surface). Measured standard deviations are of the order of 35°–40°, i.e., $\sigma_\theta \approx 0.6$ to $0.7$ rad, for which Eq. (3) yields estimates of the stream persistence length $L \approx 200$ to $280$ μm, broadly in agreement with the experimental observations (cf: Fig. 2e). Moreover, the predicted dependence $L \propto \sigma_\theta^{-2}$ provides an excellent description of the experimental behavior (Fig. 5b).

In concert with the experimental observations, the modeling results indicate that the topographic substrates used here constitute a system that differs from both locally confined and free-boundary cellular monolayers and where the collective behavior is driven not by local boundary effects as in other systems[3–5,17,18] but rather by cell activity and biased cellular orientation due to local physical constraints. In addition, while topological defect dynamics have been shown to be central in collective patterns of movement in many biological systems[17,18,36], this is not the case here because defects are expected to be essentially absent in our system due to the

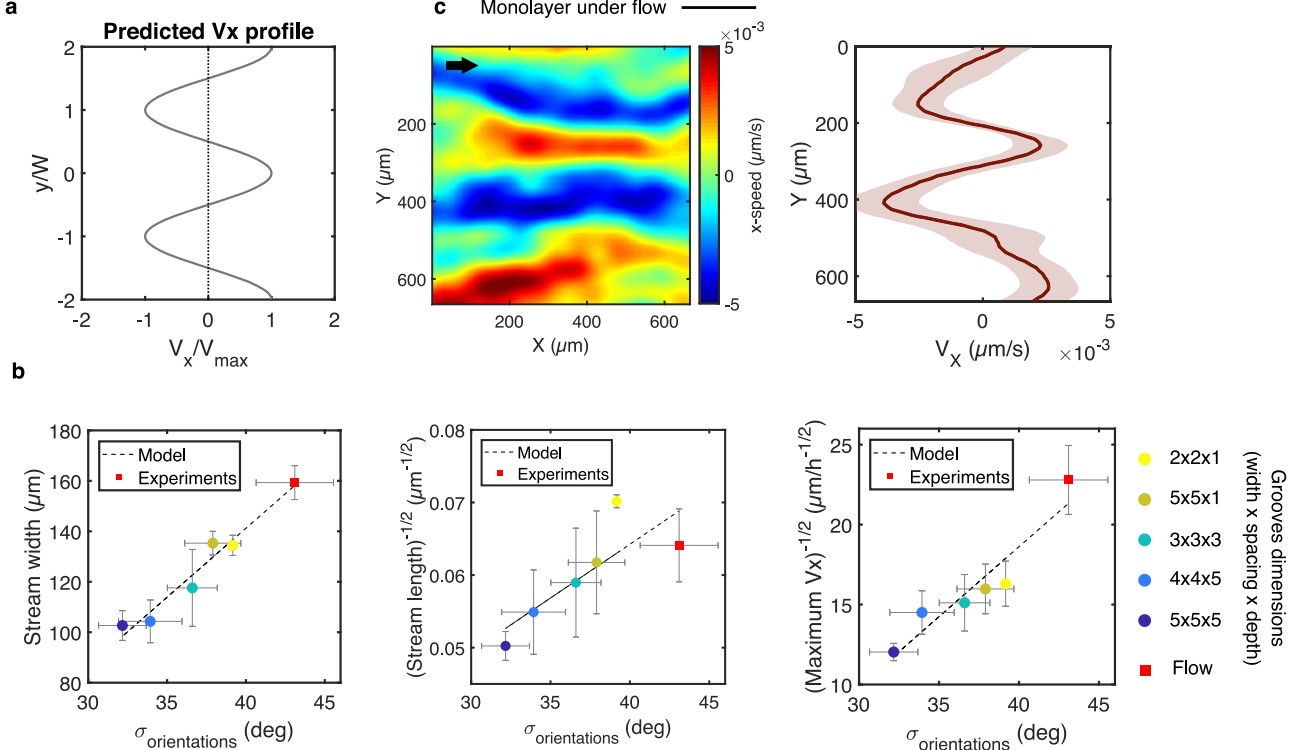

**Fig. 5 Modeling the emergence of cell streams. a** $V_x$ profile predicted by the model. **b** Predicted relationship between the spatial features or speed of the streams and the standard deviation of the displacement vector orientation angles. Dots represent experimental points and dashed lines the linear fit predicted by the model. Error bars represent SEM. **c** $V_x$ heatmap and corresponding $V_x$ profile of HUVEC monolayer on a flat surface subjected to a laminar shear stress of 2 Pa for 24 h. Black arrow indicates the flow direction. Source data are provided as a Source Data file.

homogeneous and permanent cell alignment provided by the grooves.

**External shear flow also induces antiparallel cell streams**. The model described above successfully captures the emergence of the experimentally observed cell streams and predicts that stream width and length are set by the proclivity of the cells to adopt a preferential orientation. An important consequence of this conclusion is that any source of constraint on cell orientation, and not only the anisotropic substrate topography studied thus far, would be expected to generate the periodic antiparallel stream pattern. As a proof of principle, we recorded the collective migration of HUVEC monolayers on a flat surface subjected in a parallel plate microfluidic flow chamber to steady unidirectional (x-direction) flow (shear stress of 2 Pa for 24 h). Antiparallel cell streams similar to those observed on grooved substrates were indeed observed along the flow axis direction (Fig. 5c). Remarkably, the relationship between the stream dimensions (width and length) and the cellular orientation variance follows the same scaling law predicted by our model for the topography-induced streams (red squares in Fig. 5b). We note that the cell streams induced by external flow are often less oriented along the flow axis than those induced by substrate topography. Furthermore, the streams in the direction of the applied flow tended to be associated with higher migration velocities (by ~20%) than the counter-flow streams. These differences can be attributed to the different nature of the orienting cue: in contrast to the basal grooves that do not possess a directional bias along the groove axis and that exert their effect directly on the cells' focal adhesions and cytoskeletal organization[11,13], the apical flow has intrinsic directionality and appears to have a more subtle and indirect impact on cell morphology as evidenced by the wider range of possible cell orientations (Fig. 5b). Nevertheless, the present

observations suggest that the periodic antiparallel cell streams described here may be a more universal pattern of collective movement that arises when unconfined cellular monolayers are subjected to externally orienting biophysical cues.

## Discussion
We describe here a mode of collective cell migration that takes the form of antiparallel cell streams and arises in unconfined monolayers cultured on microgroove substrates. We demonstrate that both single-cell alignment and local coordination through cell-cell contacts are necessary for the emergence of the streams and that groove depth has a considerably more pronounced impact on stream size (length and width) than groove width or spacing. The stream pattern, an example of a long-term, long-range collective pattern of movement, develops in a monolayer that exhibits minimal heterogeneities in cell activity, no sustained cell polarization, and highly dynamic cell-cell rearrangements.

An active fluid model that accounts for friction and cell orientational constraint was capable of predicting the antiparallel cell stream pattern, the dimensions of the streams, and the resulting velocity profile; thereby establishing a minimal set of physical principles sufficient to explain the emergence of the observed stream pattern. While the continuity principle, i.e. the conservation of the number of cells in the experimental chamber, imposes the presence of two opposite directions of movement, the periodic pattern of antiparallel movements arises from the non-linear dynamics of growth of an active fluid instability whose wavelength is determined by a balance among contractility, friction, and orientational constraint on cells. The predictive capability of the model was validated by experimentally modulating the level of constraint on cell orientation, which was accomplished by changing not only the groove dimensions but also by testing the effect of a different type of applied external cue,

namely unidirectional fluid flow. These findings suggest that antiparallel streams are expected to arise in any unconfined collective system where the orientation of each constitutive active element is externally constrained. In line with this idea, antiparallel streaming patterns have been reported in vitro in reconstituted microtubules aligned by a magnetic field[37], in simulated active nematic particles with high anisotropic friction[35,38], and transiently in epithelial cells on aligned collagen fibers[39]. We therefore propose that the physical mechanism described here can be viewed as a more general framework to explain streaming or laning behavior observed in other active systems.

The potential physiological relevance of the collective pattern of the movement described here remains unknown and is experimentally quite challenging to establish. In vivo, vascular endothelial cells form unconfined monolayers that are subjected to the combined effects of anisotropic guidance constraints imposed by the complex structure of the underlying basement membrane, a scenario that our microgroove substrates aimed to emulate, and directional shear forces due to luminal or interstitial blood flow. Therefore, based on the present results, we would expect the stream pattern to be present in vivo. However, ECs in vivo are much more quiescent than they are in vitro[40], and this overall quiescence appears to be associated with limited cellular motility[41], which may be expected to hinder the emergence of collective patterns of motion. Another consideration that may influence the emergence of EC streams in vivo is the curved and borderless nature of the blood vessel wall; the effects of these physical parameters remain to be elucidated.

If cell streams were to nevertheless occur in vivo, then the findings on the impact of groove depth would translate into an increased likelihood of observing these streams on thicker extracellular matrix fibers. Because loosening of arterial endothelial cell-cell junctions has been implicated in enhanced macromolecular permeability and thus in the development and progression of atherosclerotic lesions[42,43], the present finding that intact cell-cell junctions are needed for establishing the antiparallel cell stream pattern would lead us to expect this pattern to be largely absent in arterial zones prone to the development of atherosclerosis. Another consideration is the possible impact of the streams on aspects of monolayer function such as permeability regulation. Within this context, one might expect enhanced permeability in the regions at the borders between streams where cellular shearing is high compared to the zones of the streams themselves. In light of the importance of endothelial permeability regulation in normal vascular function and in the development of vascular pathologies such as atherosclerosis, this issue certainly merits further investigation.

Another area in which the results of this study may prove useful is in the design of endovascular devices such as stents or grafts where surface modification by addition of micro or nano grooves is a promising strategy for the improvement of surface endothelialization[44]. In this context, our results suggest that the dimensions of the grooves can be tailored to control the pattern of collective motion.

## Methods

**Fabrication of microgroove substrates**. The original microstructured silicon wafer was fabricated using classical photolithography procedures by UV illumination (MJB4 Mask Aligner, 23 mW/cm² power lamp, SUSS MicroTec, Germany) of a layer of SU8-2010 (MicroChem, USA) through a hard chromium mask. After exposure to vapor of trichloro(1H,1H,2H,2H-perfluorooctyl)silane (Sigma) for 20 min, the silicon wafer was used to create polydimethylsiloxane (PDMS Sylgard 184, Sigma Aldrich, ratio 1:10) replicates. To create the final coverslip on which the cells were cultured, liquid PDMS was spin coated at 1500 rpm for 30 s on the PDMS mold. Before reticulation overnight at 70 °C, a glass coverslip was placed on top of the PDMS layer. After reticulation, the glass coverslip attached to the microstructured PDMS layer was gently demolded with a scalpel and isopropanol to facilitate detachment. Microstructured coverslips were then sonicated for 10 min in ethanol for cleaning and finally rinsed in water.

**Cell culture**. Prior to cell seeding and after a 30 s plasma treatment, the microgroove substrates were incubated for 1 h with 50 µg/ml fibronectin solution (Sigma) at room temperature. The establishment of the stream pattern is not specific to fibronectin coating and was also observed with cross-linked gelatin coating (Supplementary Fig. 8b). Briefly, the coverslips were incubated with 1% gelatin for 1 h at room temperature followed by crosslinking with a 2% glutaraldehyde solution for 15 min, in accordance with the protocol described elsewhere[45]. Human umbilical vein endothelial cells (HUVECs, Lonza) in passages 4-8 were cultured in EGM2-MV medium (Lonza) at 37 °C in a humidified atmosphere of 95% air and 5% CO₂. At confluence, cells were detached with trypsin (Gibco, Thermo Fisher Scientific) and seeded onto either control PDMS substrates or microgroove coverslips at densities of 30,000–50,000 cells/cm². After 24 h of culture and prior to imaging, cells were incubated for 3 min with Hoechst 33342 (Thermo Fisher Scientific, dilution 1/10,000) for live-cell nuclear fluorescence labeling. The presence of mature cell-cell junctions at 24 h of culture was ascertained by immunostaining for different cell-cell junction proteins (Supplementary Fig. 9a). Note that the presence of cell-cell junctions rather than time in culture is the critical element for the development of the stream pattern, and streams were equally observed after 48 h of culture (Supplementary Fig. 9b).

For the ethylenediaminetetraacetic acid (EDTA, Sigma) experiments, cells were incubated after 24 h of culture with 2 or 5 mM EDTA in culture medium for the duration of the imaging protocol. Immunostaining against VE-cadherin was performed at the end of the imaging to check for EDTA treatment efficacy.

For the wound healing experiments, a 100-200 µm-wide wound was created in the monolayer perpendicular to the groove direction after 24 h of culture by gently scratching with a pipette tip. The culture was rinsed twice with PBS to remove floating cells and was fixed after 4 h.

**Immunostaining**. Culture coverslips were fixed with 4% paraformaldehyde (Thermo Fisher) in PBS for 15 min. After 1 h in a blocking solution containing 0.25% Triton and 2% bovine serum albumin (BSA), cultures were incubated for 1 h at room temperature with primary antibodies as follows: rabbit anti-VE-cadherin (ab33168, Abcam), rabbit antiphospho myosin II light chain (36671s, Cell Signaling), mouse anti-paxillin (MA5-13356, Thermofisher), mouse anti-β-catenin (C7082, Sigma Aldrich), mouse anti-ZO1 (33-9100, Thermofisher), or rabbit anti-TGN46 (ab50595, Abcam). All antibodies were diluted 1/400–1/200 in a solution containing 0.25% Triton and 1% bovine serum albumin (BSA). Coverslips were washed three times with PBS and incubated for 1 h at room temperature with Alexa Fluor 555-conjugated donkey anti-rabbit antibody (ab150074, Abcam) or Alexa Fluor 488-conjugated donkey anti-mouse antibody (ab150105, Abcam) and DAPI. F-actin staining was performed using phalloidin (Sigma).

**Time-lapse microscopy**. Live recordings of HUVEC monolayers were performed after 24 h of culture with an automated inverted microscope (Nikon Eclipse Ti) equipped with temperature and CO₂ regulation and controlled by the NIS software (Nikon). Images were acquired with a 20X objective (Nikon Plan Fluor NA=0.5) for 24 h at 15 min intervals. Three fields of views (FOVs) were randomly chosen from each experiment for the control (flat) and microgroove regions of the coverslip, and 3 to 6 separate experiments were conducted for each experimental condition.

**Flow experiments**. For flow experiments on HUVECs, $1 \times 10^5$ cells/cm² were seeded on fibronectin-coated Ibidi slides (µ-Slide I 0.4 Luer, Ibidi, Biovalley, France). After 2 h, the parallel plate flow chamber was placed on the stage of an inverted microscope (Nikon Eclipse Ti) equipped with temperature and CO₂ control and inserted into a recirculating flow loop to subject the cells to steady unidirectional shear stress. Continuous flow of cell culture medium (EGM2-MV, Lonza) was generated by a peristaltic pump (Cole-Parmer, USA). The flow exiting the pump passed through a pulse dampener (Cole-Parmer, USA) to ensure flow steadiness before reaching the cell sample. After a gradual increase of shear stress from 0.5 to 2 Pa over a period of 1 h, brightfield images were acquired using a 10X objective (Nikon Plan NA = 0.25) for 24 h at 10 min intervals. Four independent experiments were analyzed.

**Image processing and data analysis**. Tracking of the nucleus from time-lapse acquisitions was performed semi-automatically using the MTrackJ plugin in ImageJ. Subsequent analyses were performed in Matlab (MathWorks) using custom-written scripts.

*Directionality analysis*. Cell directionality was assessed using the direction autocorrelation (DA) analysis, performed by the VBA Excel Macro provided by Gorelik

et al.[46] and based on the following equations:

$$DA = \frac{1}{N-n+1} \sum_{i=0}^{N-n} \left( \vec{v}_{(i+n)\Delta t} \cdot \vec{v}_{i\Delta t} \right) = \frac{1}{N-n+1} \sum_{i=0}^{N-n} \left( \cos\left( \alpha_{(i+n)\Delta t} - \alpha_{i\Delta t} \right) \right)$$

(4)

$$\langle DA \rangle_C = \sum_{j=1}^{j=C} (DA)_j \cdot N_j / \sum_{k=1}^{k=C} N_k$$

(5)

where DA denotes the average direction autocorrelation coefficient for a given cell at step size $n$, N represents the total number of displacements, $\Delta t$ is the time interval between 2 points in the trajectory, $\alpha$ is the angle at each time point of the trajectory, and C is the total number of cells.

*Coordination analysis*. Cell-cell coordination was calculated using a method similar to that described by Hayer et al.[15] and using Eq. (6) below. More specifically, the coordination parameter $C(\Delta_R)$ was calculated as the mean cosine of the angle between the direction of one cell and that of all neighbors within a 100 μm-radius circle or 50 μm-wide circular rings located at different distances $\Delta_R$ (0–50 μm, 50–100 μm, 100–150 μm, 150–200 μm) from the considered cell. The coordination parameter is subsequently averaged for one cell along its trajectory and for all cells present in the FOV.

$$C(\Delta R) = \frac{1}{N} \sum_{i=1}^{N} \left( \frac{1}{T_i} \sum_{t=1}^{T_i} \left( \frac{1}{n_{i,t}} \sum_{j=1}^{n_{i,t}} \cos\left( \alpha_{i,t} - \alpha_{j,t} \right) \right) \right)$$

(6)

where N is the total number of cells in the FOV, $T_i$ is the total number of points in the trajectory of cell i, n is the total number of neighbors around cell i in frame t, and $\alpha_{i,t}$ is the angle of the displacement vector of cell i in frame t.

*Speed interpolation analysis*. Continuous x-speed maps were generated using a custom MATLAB script that interpolates speed values on a regular grid from the discrete positions of the nucleus extracted from the tracking.

*Cellular shear rate analysis*. Cellular shear rate in the x-direction ($SR_x$) was computed from the interpolated speed matrix as $SR_x = dV_x/dy$. Similarly, cellular y-direction shear gradient ($\nabla SR_x$) was calculated from the $SR_x$ matrix as $\nabla SR_x = dSR_x/dy$.

*Flow experiment analysis*. Particle image velocimetry (PIV) was performed on brightfield images from flow experiments using the PIVlab package in MATLAB in order to extract the x-speed matrix. The window size was set to 100 pixels = 65 μm with a 0.5 overlap for the first pass and 50 pixels = 32.5 μm for the second pass.

**Statistical analysis**. All analyses are based on 3 to 6 independent experiments using 3 different FOVs from each experiment and with each FOV containing approximately 200 to 300 cells. Statistical analyses were performed using the GraphPad Prism software. The Student *t*-test and the Mann-Whitney *t*-test were used to compare null hypotheses between two groups for normally and non-normally distributed data, respectively. Multiple groups with a normal distribution were compared by ANOVA, followed by Tukey's or Fisher's posthoc test. The number of data points for each experiment, the specific statistical tests, and the significance levels are noted in the corresponding figure legends. For the box-plots, the center line represents the median and the box limits denote the upper and lower quartiles.

## Data availability
The data that support the findings of this study are available from the corresponding author upon reasonable request. Source data are provided with this paper.

## Code availability
The codes used in analyzing the data of this study are available from the corresponding author upon reasonable request.

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

## Acknowledgements

This work was supported in part by an endowment in Cardiovascular Bioengineering from the AXA Research Fund (to AIB), a postdoctoral fellowship from the Lefoulon-Delalande Foundation (to CL), and a Biomedical Engineering Seed Grant from the Bettencourt-Schueller Foundation (to AIB). We thank Dr. Carles Blanch-Mercader for insightful discussions.

## Author contributions

C.L. and A.I.B. designed the research. C.L. and T.L. performed the experiments, C.L., A.V. and AM.D. performed data analysis, and D.G-R. developed the theory. C.L., D.G-R. and A.I.B. wrote the manuscript.

## Competing interests

The authors declare no competing interests.
