## [Peer Review File · Nature Communications]

Topography-induced large-scale antiparallel collective migration in vascular endotheliumREVIEWER COMMENTS

Reviewer #1 (Remarks to the Author):

The paper from Leclech and colleagues investigates the interaction between endothelial cells and anisotropic substrate topography upon cell confluence, leading to a harmonic pattern of collective motion. The authors apply live cell imaging to capture the movement of cell ensembles over large scales, comparing the dynamics on flat control substrates and identical ones featuring a microscopic grating, able to guide (or restrict) cell motion along the direction dictated by the topography. Characteristic patterns of antiparallel flow arise, generated by streams of cells that move against each other. The spatial and temporal organization of these patterns depends on the establishment and maturity of the cell-to-cell junctions, as they are largely ablated by their disassembly. In addition, other anisotropic signal such as unidirectional fluid flow, similarly induce the formation of the antiparallel streams. Finally, the authors propose a model that capture the spatial organization as experimentally described.

Altogether the work provides an elegant description of an interesting phenomenon, which is novel and has not been described to this level of detail in the contest of collective cell activities. However, the results are purely observational, and the model adds little to the understanding of this complex behavior. The manuscript shall become acceptable for publication in Nat Com after major revisions as noted below:

Main:

1. How – From the data presented it is now really clear how the antiparallel streams emerge, in particular if it is an all-active process or only partially propelled by groups of cells which generate motion, and others that are passively dragged to make space. A possible answer could come from traction force microscopy, which could be applied to the experiments under flow, where a flat PDMS substrate is used.
2. How – In line with what proposed above, another indication on the active/passive nature of the movement shall come from the analysis of cell polarization. Endothelial cells acquire planar cell polarization (PCP) in response to anisotropic stimuli (typically flow) and move the Golgi upstream of the nucleus. Several papers describe this process, which can be visualized adding a live Golgi maker. Polarization vectors shall tell if cells are performing directional migration.
3. Why – Why do the antiparallel streams emerge? What is their function? Are they improving the performance of wound healing? All experiments provided seem to be performed with confluent monolayers, so a front advancement cannot be measured. Can the model predict how the two conditions would influence the closure of an open space?
4. Technical - I have a few doubts on how the experiments were performed.
 - a. 24 h incubation time seems to be short for HUVEC to form mature junctions. Please check the seminal paper from Lampugnani (J Cell Biology – 1997) to correctly tune density and incubation time.
 - b. Fibronectin coating is surely sub-optimal. Junction formation is supported by cross linked gelatin, which should also be a more stable coating under flow. Again the early work from the group of Prof. Dejana should be taken as reference.

Minor:

1. Coating homogeneity should be demonstrated on gratings. Authors should use fluorescent FN (or better gelatin) to show that their protocol achieves homogeneous coating on groves and ridges, and exclude that the effect they see is purely or partially haptotactical.
2. The effect of gratings should be better described. A number of reports have shown the distribution of focal adhesions on the top of ridges, however it would be useful to confirm that the substrates used here have the same effect.

Reviewer #2 (Remarks to the Author):

The manuscript of Leclech et al presents an experimental and theoretical study of collective migration of endothelial vascular cells on a PDMS substrate that has been patterned by parallel grooves. The experimental results are spectacular as they show periodic antiparallel streams of flowing cells. To my knowledge, these results are new and original. They are interpreted by a theoretical model based on the work of Duclos et al. that describes flow instabilities in active nematic stripes based on active gel theory. I believe that the manuscript cannot be published in Nature Communication under its present form and that the authors should address before the following comments and questions.

-Should the cells be considered as nematic or polar? The experiments strongly suggest that they are polar but the theory treats them as a nematic phase as it is invariant by changing \mathbf{p} to $-\mathbf{p}$. To me, this is an important issue in the choice of the theory that here considers nematic order.

-Is the cell density homogeneous or does it also have a periodic variation perpendicular to the flow?

-The friction term in equation 4 is not discussed. It includes a passive friction on the substrate (ξ) and an active term proportional to the velocity v_0 . Why is the active term proportional to the unit vector along the local velocity. A more natural choice (in the spirit of the active gel theory where the only vector of the theory is the polarisation \mathbf{p}) would be to write that the velocity is proportional to the polarisation \mathbf{p} . This is what has been done in Ref 23 by Blanch Mercader and coworkers. Can this theory also be used to interpret the data?

-The authors introduce a nematic field α to take into account the effect of the grooves and they show experimentally that this field mostly depends on the depth of the grooves. They then estimate this field using linear response theory but for an equilibrium system. I am not sure how to justify this estimation, the system is here strongly out of equilibrium and it is not obvious to me that the effective temperature $D \xi$ can be used to calculate the fluctuations of the orientation.

-The linear stability analysis is clear and indeed predicts periodic states with a period that is independent of the spontaneous velocity v_0 . The calculation of the period is then done by minimizing the free energy. The reason why the minimization of the free energy for this non equilibrium system provides the period is not clear to me and should be explained. The standard way to perform the non linear calculation of the velocity profile would be to expand systematically to next order (I guess third order) and then to expand the velocity in Fourier modes; Would that give the same result. It appears also that the period is very different than the fastest growing mode obtained by the linear stability analysis. Is there a reason for that?

-The solution for the velocity profile is obtained independently over the two half periods. I guess that there is an interface between the two half periods and that the relevant length scale for the width of the interface could be $(K/\alpha)^{1/2}$

Responses to Reviewers

We would like to thank the reviewers for their insightful and constructive comments. We have endeavored to respond positively to each of their concerns, and we believe that the resulting changes have yielded a more complete and much improved manuscript. Below is a point-by-point response to the specific points raised by the referees. The reviewers' comments are shown in italics. All changes made in the revised version of the manuscript are visible in Track Changes mode. Page and figure numbers cited below refer to the revised manuscript.

Reviewer 1

Altogether the work provides an elegant description of an interesting phenomenon, which is novel and has not been described to this level of detail in the context of collective cell activities. However, the results are purely observational, and the model adds little to the understanding of this complex behavior. The manuscript shall become acceptable for publication in Nat Com after major revisions as noted below:

Major:

1. How – From the data presented it is not really clear how the antiparallel streams emerge, in particular if it is an all-active process or only partially propelled by groups of cells which generate motion, and others that are passively dragged to make space. A possible answer could come from traction force microscopy, which could be applied to the experiments under flow, where a flat PDMS substrate is used.

The reviewer raises a very interesting question that led us to perform a number of additional experiments and to enrich our vision of the results. The new experiments and analyses performed to explore this point have been grouped in a new section in Results entitled “The emergence of the streams is not associated with large heterogeneities in cell activity”, an associated new figure in the main text (Fig. 4 in the revised manuscript), and three additional figures (Figs. 5, 6, and 7) in the Supplementary Information section.

In light of the reviewer’s question, different configurations of cell activity within the monolayer can be considered to explain the emergence of the cell streams. A first situation, which is the one assumed in the modeling, is an “all-active” system, where all the cells are considered equivalent in terms of activity (migration, traction force generation, contractility, polarization, etc.) and participate equally in the emergence of the pattern of movement. A second possibility, highlighted in the reviewer’s question and referred to here as the “partially-active” scenario, is a situation where heterogeneities in cell activity exist within the monolayer and contribute to driving the emergence of the streams. This scenario would then posit the presence of groups of cells that actively drive the movement of other less active cells, a situation reminiscent of the leader-follower system present in wound healing, for instance. To try to distinguish between these two possibilities, we considered different implications of the partially-active scenario.

One expected consequence of the partially-active scenario is, as pointed out by the reviewer, the existence of spatial heterogeneities in cellular traction forces. This might lead, for example, to clusters in the traction force maps, corresponding to the presence of actively pulling groups of cells within the streams. Although traction force microscopy (TFM) can in theory be used to address this point, this approach remains very challenging both experimentally and in the interpretation of its results. A number of previous studies have

already reported TFM measurements on endothelial cells under flow ¹⁻³. These studies yielded different and sometimes conflicting results, due to both the large variability and the spatial and temporal heterogeneity reported in the traction force maps. The difficulty of performing and interpreting the results of these experiments, in addition to the absence of regular and coherent traction force patterns in those studies, led us to conclude that this type of experiment would not help us to convincingly address the concern raised by the reviewer. Instead, we have chosen to explore a number of other implications of the partially-active “leader-follower-type” hypothesis, and the results have provided us with valuable insight as described next.

Cellular heterogeneities in actomyosin contractility and focal adhesion morphology:

The generation of traction forces is mediated by focal adhesions (FAs) and actomyosin-generated contractility. Differences in cell activity and traction forces are therefore expected to translate into different levels and organization of the actomyosin network ⁴⁻⁶ as well as and different sizes of FAs since FA size correlates with traction force ^{7,8}. Immunostaining for actin and phospho myosin light chain (pMLC) in monolayers on 5 μm -deep grooves revealed that cells exhibit a broadly homogeneous organization of F-actin, forming dense and regular stress fibers along the grooves and throughout the cell (Fig. 4a). The levels of actin and associated pMLC were fairly constant across the monolayer, as quantified by the ratio of normalized fluorescence intensity between individual cells and their neighbors, which fluctuated around 1 (Fig. 4a). Similar results were obtained by measuring the area of FAs, which did not vary considerably within the monolayer (Supplementary Fig. 5). These new results suggest that there is no significant spatial heterogeneity in either contractility or traction forces among cells within the monolayer.

Heterogeneities in cell-cell junction morphology: Another manifestation of cell traction forces within a monolayer is an effect on the morphology of cell-cell junctions. In vascular endothelium, previous studies have described mature linear junctions vs. “zipper-like” or “finger-like” junctions that are under tension and that are particularly present between leader and follower cells ^{6,9,10}. We analyzed cell-cell junction morphology in monolayers on 5 μm -deep grooves by immunostaining for different proteins (β -catenin or VE-cadherin) and quantified junction linearity by calculating the directionality ratio (i.e. the tortuosity) associated with the outline of the junctions. We observed that while linear junctions were present in the groove direction, zipper-like junctions were present perpendicular to the groove and stream directions, confirming the general direction of traction in the monolayer. However, the morphologies of the zipper-like junctions were similar across the monolayer, suggesting minimal heterogeneity in traction forces among cells. These results are shown in Supplementary Fig. 6a.

Heterogeneities in cell elongation: Actively pulling leader cells have been shown to be more elongated than follower cells ¹¹. We therefore analyzed cell shape in monolayers on 5 μm -deep grooves by automatic segmentation of cell outlines from cell-cell junction staining. Although cells within the monolayer exhibited different levels of elongation, as quantified by the cell circularity (minor-to-major axis ratio), no clustering or regular pattern of elongation was present. In addition, dynamic analysis of cell shape demonstrated that a single cell within the monolayer can, in the span of hours, explore the entire range of instantaneous cell elongations found within the monolayer, suggesting rapidly evolving patterns of cellular forces. These results are shown in Supplementary Fig. 6b.

Dynamics of cell-cell rearrangement: From a dynamic point of view, the presence of groups of leading cells in the streams would be expected to be associated with long-term cohesiveness among the cells within the monolayer and limited cellular rearrangement, at least between leaders and their followers. We quantified the incidence of cell-cell rearrangements by tracking the evolution of each cell's neighbors (defined as cells within a 100 μm -radius) during the entire recording period. Parameters such as the mean time spent with the same neighbor and the rate of neighbor exchange (defined as the frequency of gain or loss of neighbors) extracted from this analysis revealed that cells within the monolayer have highly dynamic or "fluid" neighborhoods with frequent cellular rearrangements. This fluidity is confirmed by the observations of cells changing direction within or between streams, counter-stream migrating cells, and cells overtaking one another within a stream. In addition, quantification of cell-cell rearrangements showed similar values for cells either within streams or at the border between streams, and these values were close to those obtained for cells in monolayers on flat surfaces. This suggests that if leader-follower-type patterns are present, they would be expected to be very local and highly transient. These findings, which are shown in Fig. 4c, point to the interesting notion that long-term coordination is not necessary for the emergence of the streams.

Propagation of streams beyond grooves: If a group of active cells pulls a group of passive cells in forming the cell streams, then one might expect a level of propagation of this effect beyond the microgrooves. To test this idea, we imaged the monolayer at the border between grooved and flat regions of the substrate. Consistent with the frequent cellular rearrangements described above, no propagation of the streams or of cellular alignment and elongation present on the microgrooves was observed on the flat region. Interestingly, this plasticity of ECs was visible even at the subcellular level where two clearly distinct patterns of F-actin organization and orientation are observed in cells that are positioned partly on grooves and partly on flat surfaces. These results are shown in Supplementary Fig. 7.

2. How – In line with what proposed above, another indication on the active/passive nature of the movement shall come from the analysis of cell polarization. Endothelial cells acquire planar cell polarization (PCP) in response to anisotropic stimuli (typically flow) and move the Golgi upstream of the nucleus. Several papers describe this process, which can be visualized adding a live Golgi maker. Polarization vectors shall tell if cells are performing directional migration.

We thank the reviewer for this suggestion. Another prominent feature of leading cells in collective migration is indeed cell polarization^{12,13}. To explore this point and as suggested by the reviewer, we analyzed the position of the Golgi apparatus with respect to the nucleus by staining for the trans-Golgi network protein TGN46 in endothelial monolayers. On both flat and microgroove substrates, the Golgi remained very close to the nucleus and did not exhibit any preferential localization around the nucleus, as assessed by the random orientations of the polarization vectors. As "positive" controls, we measured the polarization in endothelial cells under flow and in a wound healing assay on microgrooves. In the flow experiments, as already reported^{14,15} and mentioned by the reviewer, we observed clear preferential polarization of the Golgi upstream of the nucleus. In the wound healing experiments, endothelial cells at the edge of the wound polarized towards the wound with their Golgi in front of the nucleus, in the direction of movement¹⁶. These results are shown in Fig. 4b. The absence of clear polarization in the monolayers on grooves is in line with the high level of

cellular homogeneity, plasticity, and fluidity described above. Interestingly, these findings also suggest that an orientational cue that has no polarity, as is the case with the grooves, is insufficient to polarize endothelial cells, particularly in monolayers where they have no significant intrinsic polarization. Additionally, the results show that cell polarization is not necessary for stream emergence.

To conclude, all of the new findings presented above argue against the presence of a stable leader-follower system with groups of cells actively driving the streams by pulling groups of less active cells along. Rather, viewed globally, our observations point toward a high level of cellular homogeneity in terms of polarization, contractility, and traction forces within endothelial cell monolayers, with all cells actively participating in the antiparallel cell streams. Of course, leader-follower-type interactions among cells cannot be completely excluded, but our results suggest that were such interactions to occur, they would do so on small spatial and temporal scales, much smaller than the characteristic scales associated with the streams. These results and associated discussion have been incorporated into the revised manuscript. We believe that in addition to providing additional insight into the mechanism of stream emergence, these new results also underscore interesting information on cellular and dynamic intrinsic properties of vascular endothelial monolayers. They also reinforce the pertinence of the physics-based modeling section which inherently assumes that the cellular monolayer is an all-active system.

3. Why – Why do the antiparallel streams emerge? What is their function? Are they improving the performance of wound healing? All experiments provided seem to be performed with confluent monolayers, so a front advancement cannot be measured. Can the model predict how the two conditions would influence the closure of an open space?

In terms of why the streams emerge, we believe that the explanation is provided by the modeling part of the study. The main conclusion of the model is that antiparallel cell streams are expected to emerge in unconfined cell monolayers where each cell is oriented externally, with the dimensions of the streams being determined by the intensity of cell alignment. These predictions were validated experimentally by changing either the groove dimensions or the type of external cue applied (grooves vs. flow), thereby eliciting different degrees of constraint on cell orientation. Consequently, as now detailed in the Discussion section of the revised manuscript, we propose that the mechanism of stream formation described here can be viewed as a more general framework that also explains the streaming or laning behavior observed in other systems^{17–20}. We hope that the additional discussion we have now included better clarifies the aim and interest of the model and its link to the experiments.

As for the function of the streams, that remains a more elusive question for which we can only provide elements of discussion at this point. The reviewer raises the intriguing possibility that the streams may improve the wound healing performance of the monolayer. Unfortunately, the model cannot be used to address this question because this would require the model to include the description of a “free surface”, thus introducing new physical parameters (such as an “effective line tension”) which would then need to be estimated from wound closure experiments, somewhat defeating the purpose. Moreover, the partial differential equations, which are already rather complex (see the new nonlinear stability analysis in the Supplementary Note), would then depend on an additional variable (two spatial coordinates instead of one). This would significantly complicate the model’s

interpretation and likely preclude the analytical treatment of the instability that we succeeded to perform here. We nevertheless tried to explore this question experimentally by performing wound healing assays on both flat surfaces and 5 μm -deep microgrooves (wound edge perpendicular to the grooves), with or without EDTA to distinguish the effect of streams from the grooves themselves (recall that EDTA eliminates the streams). The results of the experiments are shown in the figure below. The first important observation is that the streams completely disappear in the vicinity of the wound, rendering their potential involvement in modulating wound closure rates moot. By quantifying the movement of individual cells, we were able to establish that while grooves promoted more rapid cell migration relative to flat surfaces shortly after wounding, the final wound closure rates were not different in the two cases. These results suggest that the streams do not play a role in improving wound healing. As these experiments are far from the main focus of this study, we decided not to include them in the revised version of the manuscript. We would be happy to add them if the reviewer deems it necessary.

Figure: Wound healing experiments on microgrooves. **a**, Accumulated cell trajectories color-coded by the orientation of the displacement at early time points (0 - 7.5 h after wounding) and at later times (7.5 - 14 h). The dashed vertical lines indicate the initial edges of the wound. **b**, Quantification of early and late wound healing based on 3 independent experiments. One-way ANOVA, Tukey's post-test (*, $p < 0.05$; **, $p < 0.005$).

As far as other possible functions of the streams are concerned, different elements are interesting to discuss. A first point is whether or not the streams might be expected to be present *in vivo*. As endothelial cells are often aligned and elongated in blood vessels due to a combination of cues including anisotropic constraints imposed by the structure of the underlying basement membrane and the directional shear forces due to luminal blood flow, a streaming behavior could in theory be expected *in vivo*. However, endothelial cells *in vivo* are

much more quiescent than they are *in vitro*. For instance, the average lifespan of an endothelial cell *in vivo* is on the order of one year²¹. This overall quiescence *in vivo* is likely to be associated with limited cellular motility. Indeed, although the literature on endothelial cells migration speeds *in vivo* is very limited, the available data suggest greatly reduced motility relative to the *in vitro* environment. For instance, corneal endothelial cells have been reported to migrate at a mean velocity of around 16 $\mu\text{m}/\text{day}$ ²². This can be compared to the velocities of 20-30 $\mu\text{m}/\text{h}$ observed in the current study. The high level of quiescence and limited motility of endothelial cells *in vivo* may be expected to hinder the emergence of collective patterns of motion. Another consideration that may influence the emergence of endothelial cell streams *in vivo* is the continuous and curved nature of the blood vessel wall. If streams were to nevertheless be present *in vivo*, this collective pattern of movement may be expected to have an impact on aspects of monolayer function such as permeability regulation. Within this context, one might expect enhanced permeability in the regions at the borders between streams where cellular shearing is high compared to the zones of the streams themselves. In light of the importance of endothelial permeability regulation in normal vascular function and in the development of vascular pathologies such as atherosclerosis, this issue certainly merits further investigation.

Another area in which the results of this study may prove useful is in the design of endovascular devices such as stents or grafts where surface modification by addition of micro or nano grooves is a promising strategy for the improvement of surface endothelialization²³. In this context, our results suggest that the dimensions of the grooves can be tailored to control the pattern of collective motion.

All of the arguments about the potential functions of the streams described above have been added to the Discussion section of the revised manuscript.

4. Technical - I have a few doubts on how the experiments were performed.

a. 24 h incubation time seems to be short for HUVEC to form mature junctions. Please check the seminal paper from Lampugnani (J Cell Biology – 1997) to correctly tune density and incubation time.

As the reviewer points out, tuning the initial cell seeding density allows attaining confluence after different times in culture. The seeding density used here (30,000 – 50,000 cells/cm²) was optimized so that a confluent monolayer where cell-cell junctions are present is obtained after 24 h of culture. We note that these values are consistent with the studies of Lampugnani *et al.* (Lampugnani *et al.*, 1997, 1995) where much lower seeding densities (4,000 – 10,000 cells/cm²) were used with the goal of reaching confluence after 72-96 h of culture.

To ascertain the presence of cell-cell junctions in our monolayers after 24 h of culture, we have complemented the VE-cadherin staining images shown in Fig. 1 of the original submission with additional staining for the junctional proteins β -catenin and ZO-1 which also showed the presence of clear junctions. In particular, the presence of the tight junction protein ZO-1 indicates that cells have established relatively mature and functional junctions. As another piece of evidence, we recorded the movement of endothelial cell monolayers on grooves after 48 h of culture and observed stream patterns similar to those seen with 24 h cultures, thereby indicating that the time of incubation (and associated maturation of

junctions) does not influence this collective pattern of movement. These new results are shown in Supplementary Fig. 9 of the revised manuscript.

Another important point is that as we had shown in Fig. 3 and Supplementary Fig. 4, the streams actually begin to be visible at subconfluence when the cells start to form junctions, become more clearly established for confluent monolayers (equivalent to the “recently confluent” case in the studies of Lampugnani), but are no longer visible in highly confluent monolayers (equivalent to the “long-confluent” case in the studies of Lampugnani) due to a strong decrease in cellular motility (reminiscent of a jamming phenomenon). These observations show that while cell-cell contacts are necessary for the establishment of the streams, it is likely that the maturation of junctions associated with longer times in culture/higher cell densities does not influence the emergence of the streams (and would rather tend to dampen them). Therefore, it seems unnecessary in the context of the present study to increase the time in culture to target more mature junctions. These various elements are now briefly mentioned in the section associated with Fig. 3 of the revised manuscript, and examples of junction morphologies for different monolayer densities are provided in Supplementary Fig. 4.

b. Fibronectin coating is surely sub-optimal. Junction formation is supported by cross linked gelatin, which should also be a more stable coating under flow. Again the early work from the group of Prof. Dejana should be taken as reference.

As the reviewer undoubtedly knows, fibronectin coating is widely used in the endothelial cell culture literature, including in the early studies of Prof. Dejana mentioned by the reviewer^{26,27}. Fibronectin coating has been reported to be as effective (in terms of adhesion and proliferation at least) as other types of coating such as collagen, laminin, or gelatin^{28–30}. In our hands, endothelial cells cultured on fibronectin are always able to form monolayers with mature junctions and are highly stable under flow.

It is true, however, that the potential influence of the coating protein on endothelial cell collective migration patterns has not been studied and is an interesting question. To address the reviewer’s concern, we performed recordings of endothelial cell monolayers on microgroove substrates coated with cross-linked gelatin as suggested. We observed the presence of prominent and stable antiparallel cell streams with dimensions similar to those observed with fibronectin coating. While these results show that cross-linked gelatin is indeed an effective coating for promoting the emergence of the streams, they also indicate that the type of coating does not influence the establishment of the pattern of movement described in this study. This information has been added to the Methods section and Supplementary Fig. 8b of the revised manuscript.

Minor:

1. Coating homogeneity should be demonstrated on gratings. Authors should use fluorescent FN (or better gelatin) to show that their protocol achieves homogeneous coating on groves and ridges, and exclude that the effect they see is purely or partially haptotactical.

To address this issue, we used fluorescent fibrinogen (Thermofisher F35200) mixed with fibronectin to visualize coating homogeneity. 3D reconstruction and cross-sections from

confocal microscopy stacks showed a homogeneous coating on all groove surfaces. This result is shown in the new Supplementary Fig. 8a of the revised manuscript.

2. The effect of gratings should be better described. A number of reports have shown the distribution of focal adhesions on the top of ridges, however it would be useful to confirm that the substrates used here have the same effect.

The reviewer brings up an interesting point. Focal adhesions (FAs) are indeed important and have been proposed to be central players in cell alignment and elongation on grooves³¹. A number of studies have investigated the localization and morphology of FAs on these surfaces. In our system, FAs are indeed observed on ridges, and especially at the ridge edges. In addition, we observe that FAs are more aligned and elongated on microgrooves compared to a flat surface, as has been reported in different cell types^{32,33}. A more precise description of the effect of grooves on ECs has been added to the beginning of the Results section and in the new Supplementary Fig. 1.

Reviewer 2

The manuscript of Leclech et al presents an experimental and theoretical study of collective migration of endothelial vascular cells on a PDMS substrate that has been patterned by parallel grooves. The experimental results are spectacular as they show periodic antiparallel streams of flowing cells. To my knowledge, these results are new and original. They are interpreted by a theoretical model based on the work of Duclos et al. that describes flow instabilities in active nematic stripes based on active gel theory. I believe that the manuscript cannot be published in Nature Communication under its present form and that the authors should address before the following comments and questions.

We thank the reviewer for his/her appreciation of our experiments and for valuable insights that led us to significantly improve our theoretical model as discussed below.

- Should the cells be considered as nematic or polar? The experiments strongly suggest that they are polar but the theory treats them as a nematic phase as it is invariant by changing \mathbf{p} to $-\mathbf{p}$. To me, this is an important issue in the choice of the theory that here considers nematic order.

The choice of describing cells as nematic or polar is indeed not obvious, and it depends on the cell type. Our choice of a nematic description is consistent with previous nematic descriptions of similar systems³⁴ and based on the following experimental observations: (1) the cells used here do not exhibit any marked front-to-back structural asymmetry, associated with a low intrinsic polarization (as is now shown in Fig. 4b), and (2) cells can spontaneously reverse their direction of motion without turning the cell body over time scales smaller than the time scale of stream development. This important point is now discussed in the revised manuscript as well as in the Supplementary Note. A new supplementary video 4 has also been added to illustrate polarity reversal events.

- Is the cell density homogeneous or does it also have a periodic variation perpendicular to the flow?

The cell density is homogeneous and does not vary perpendicular to the flow, as is shown below by quantification of the density of cell nuclei as a function of the distance from the stream centerlines. This observation is consistent with our description of the system as nematic, since polar terms usually yield spatial inhomogeneities in the concentration that are typically not observed in nematic systems³⁵.

Figure: Homogeneity of cell density within streams. Left: Probability of encountering a cell nucleus as a function of the normalized absolute distance from the stream centerline (0 = stream border, 100 = stream centerline). Each dot represents a field of view from 3 independent experiments. Right: frame extracted from a recording illustrating the spatially homogeneous distribution of nuclei (blue) across the field of view and the streams.

- The friction term in equation 4 is not discussed. It includes a passive friction on the substrate (ξ) and an active term proportional to the velocity v_0 . Why is the active term proportional to the unit vector along the local velocity. A more natural choice (in the spirit of the active gel theory where the only vector of the theory is the polarisation \mathbf{p}) would be to write that the velocity is proportional to the polarisation $\mathbf{v} \propto \mathbf{p}$. This is what has been done in Ref 23 by Blanch Mercader and coworkers. Can this theory also be used to interpret the data?

The alternative formulation of this active term proposed by the reviewer can indeed be used. However, as the reviewer pointed out in another comment below, the predicted stream width and length are independent of the spontaneous velocity, which turns out not to be a necessary model ingredient to predict the emergence of the streams. Because our aim is to propose the simplest model that can explain the observations, we have revised the model to omit this unnecessary active term, v_0 . Importantly, the revised analysis yields an expression for the maximum stream velocity, which is now a model prediction rather than an imposed parameter. These changes are reflected in the revised manuscript (new Fig. 5) and Supplementary Note.

- The authors introduce a nematic field α to take into account the effect of the grooves and they show experimentally that this field mostly depends on the depth of the grooves. They then estimate this field using linear response theory but for an equilibrium system. I am not sure how to justify this estimation, the system is here strongly out of equilibrium and it is not obvious to me that the effective temperature $D \xi$ can be used to calculate the fluctuations of the orientation.

We describe cell misalignment as an energetic cost as this provides a reasonable and quantitative prediction of the experimental cell behavior over different groove geometries. The reviewer is correct in that the system is out of equilibrium, so our use of equilibrium terminology was inaccurate. As is customarily done in the description of such active systems, we now refer to the energetic quantities as *effective energies* and to the observed profiles as *steady-state* profiles rather than *equilibrium* profiles. We note that the effective free energy in the most general active gel theory also includes an active alignment term³⁶, which is analogous to our parameter α . Moreover, while the Stokes-Einstein relation we use to quantify the field α is formally valid for equilibrium systems, some theoretical work has investigated its extension to certain kinds of active systems, through the concept of an *effective temperature*³⁷. Specifically, it has been shown that the Stokes-Einstein relation is applicable to certain active systems, notably in the dilute limit³⁸. For this reason, we have performed new quantifications of the standard deviation of cell orientations and thereby the parameter α using very low cell densities, which yield similar values as our previous estimations in monolayers. This point is now discussed in the Supplementary Note.

- The linear stability analysis is clear and indeed predicts periodic states with a period that is independent of the spontaneous velocity v_0 . The calculation of the period is then done by minimizing the free energy. The reason why the minimization of the free energy for this non equilibrium system provides the period is not clear to me and should be explained. The standard way to perform the non linear calculation of the velocity profile would be to expand systematically to next order (I guess third order) and then to expand the velocity in Fourier mode. Would that give the same result. It appears also that the period is very different than the fastest growing mode obtained by the linear stability analysis. Is there a reason for that?

We thank the reviewer for this important remark that has led us to a major improvement of our theoretical description. The first-order linear stability analysis yields a sinusoidal velocity profile, and it predicts a range of possible wavelengths, among which is one that corresponds to the fastest-growing mode. As suggested by the reviewer, we have performed the next-order (third-order) analysis, which includes corrections to the first and third harmonics. These corrections are however small (two orders of magnitude smaller than the leading term) and thus negligible in the comparison with the experiments. However, we cannot conclude that the fastest-growing mode predicted by the linear analysis corresponds to the observed wavelength of the streams, as the experimental system may be far into the nonlinear regime. Therefore, we have conducted a new, non-linear stability analysis by using a method first proposed by Sharma & Ruckenstein³⁹, as detailed in the revised Supplementary Note. This nonlinear analysis shows that as the perturbation grows, the wavelength of the fastest-growing mode decreases, thus suggesting that the wavelength of the fully developed streams corresponds to the minimum instability wavelength, at which the growth rate becomes zero and the system reaches steady-state. Interestingly, this minimum instability wavelength corresponds to the stream width that we previously obtained by minimizing a free

energy, an approach that we have now omitted from the revised manuscript. This prediction of the stream width, now deduced from the stability analysis, is in good agreement with the experimental observations. Moreover, the revised approach allows predicting the dependence of the stream velocity on the parameter α . The manuscript, Supplementary Note, and new Fig. 5 have been significantly revised to reflect the improved theoretical approach.

- The solution for the velocity profile is obtained independently over the two half periods. I guess that there is an interface between the two half periods and that the relevant length scale for the width of the interface could be $(K/\alpha)^{1/2}$

The reviewer is right. However, in our revised description of the velocity profile, which is based on a non-linear stability analysis, we no longer give a solution over two half periods, so discussion of this interface is no longer relevant.

References

1. Hur, S. S. *et al.* Roles of cell confluency and fluid shear in 3-dimensional intracellular forces in endothelial cells. **109**, (2012).
2. Perrault, C. M. *et al.* Traction Forces of Endothelial Cells under Slow Shear Flow. *Biophys. J.* **109**, 1533–1536 (2015).
3. Steward, R. *et al.* Fluid shear, intercellular stress, and endothelial cell alignment. *Am. J. Physiol. Cell Physiol.* **308**, C657-64 (2015).
4. Kollimada, S. *et al.* The biochemical composition of the actomyosin network sets the magnitude of cellular traction forces. *Mol. Biol. Cell* **32**, 1737–1748 (2021).
5. Pandya, P., Orgaz, J. L. & Sanz-Moreno, V. Actomyosin contractility and collective migration: may the force be with you. *Curr. Opin. Cell Biol.* **48**, 87–96 (2017).
6. Hayer, A. *et al.* Engulfed cadherin fingers are polarized junctional structures between collectively migrating endothelial cells. *Nat. Cell Biol.* **18**, 1311–1323 (2016).
7. Trichet, L. *et al.* Evidence of a large-scale mechanosensing mechanism for cellular adaptation to substrate stiffness. *Proc. Natl. Acad. Sci.* **109**, 6933–6938 (2012).
8. Balaban, N. Q. *et al.* Force and focal adhesion assembly: a close relationship studied using elastic micropatterned substrates. *Nat. Cell Biol.* **3**, 466–472 (2001).
9. Huveneers, S. *et al.* Vinculin associates with endothelial VE-cadherin junctions to control force-dependent remodeling. *J. Cell Biol.* **196**, 641–652 (2012).
10. Yang, Y. *et al.* Probing Leader Cells in Endothelial Collective Migration by Plasma Lithography Geometric Confinement. *Sci. Rep.* **6**, 22707 (2016).
11. Vishwakarma, M. *et al.* Mechanical interactions among followers determine the emergence of leaders in migrating epithelial cell collectives. *Nat. Commun.* **9**, (2018).
12. Mayor, R. & Etienne-Manneville, S. The front and rear of collective cell migration. *Nat. Rev. Mol. Cell Biol.* **17**, 97–109 (2016).
13. Khalil, A. A. & Friedl, P. Determinants of leader cells in collective cell migration. *Integr. Biol.* **2**, 568 (2010).
14. Tkachenko, E. *et al.* The nucleus of endothelial cell as a sensor of blood flow direction. *Biol. Open* **2**, 1007–1012 (2013).
15. Morgan, J. T. *et al.* Integration of basal topographic cues and apical shear stress in vascular endothelial cells. *Biomaterials* **33**, 4126–35 (2012).
16. Gotlieb, A. I., May, L. M., Subrahmanyam, L. & Kalnins, V. I. Distribution of microtubule organizing centers in migrating sheets of endothelial cells. *J. Cell Biol.* **91**, 589–594 (1981).
17. Thampi, S. P., Golestanian, R. & Yeomans, J. M. Active nematic materials with substrate friction. *Phys. Rev. E - Stat. Nonlinear, Soft Matter Phys.* **90**, (2014).
18. Thijssen, K., Metselaar, L., Yeomans, J. M. & Doostmohammadi, A. Active nematics with anisotropic friction: The decisive role of the flow aligning parameter. *Soft Matter* **16**, 2065–2074 (2020).
19. Haga, H., Irahara, C., Kobayashi, R., Nakagaki, T. & Kawabata, K. Collective movement of epithelial cells on a collagen gel substrate. *Biophys. J.* **88**, 2250–2256 (2005).

20. Guillaumat, P., Ignés-Mullol, J. & Sagués, F. Control of active liquid crystals with a magnetic field. *Proc. Natl. Acad. Sci.* **113**, (2016).
21. Montezano, A. C., Neves, K. B., Lopes, R. A. M. & Rios, F. Isolation and Culture of Endothelial Cells from Large Vessels. in 345–348 (2017). doi:10.1007/978-1-4939-6625-7_26
22. Correll, M. H. *et al.* In Vivo Labeling and Tracking of Proliferating Corneal Endothelial Cells by 5-Ethynyl-2'-Deoxyuridine in Rabbits. *Transl. Vis. Sci. Technol.* **10**, 7 (2021).
23. Tan, C. H., Muhamad, N. & Abdullah, M. M. A. B. Surface Topographical Modification of Coronary Stent: A Review. in *IOP Conference Series: Materials Science and Engineering* **209**, (Institute of Physics Publishing, 2017).
24. Lampugnani, M. G. *et al.* The molecular organization of endothelial cell to cell junctions: differential association of plakoglobin, beta-catenin, and alpha-catenin with vascular endothelial cadherin (VE-cadherin). *J. Cell Biol.* **129**, 203–217 (1995).
25. Lampugnani, M. G. *et al.* Cell confluence regulates tyrosine phosphorylation of adherens junction components in endothelial cells. *J. Cell Sci.* **110 (Pt 17)**, 2065–77 (1997).
26. Dejana, E. *et al.* Fibronectin and vitronectin regulate the organization of their respective Arg-Gly-Asp adhesion receptors in cultured human endothelial cells. *J. Cell Biol.* **107**, 1215–1223 (1988).
27. Dejana, E. *et al.* Fibrinogen induces adhesion, spreading, and microfilament organization of human endothelial cells in vitro. *J. Cell Biol.* **104**, 1403–1411 (1987).
28. Smeets, E. F., von Asmuth, E. J. U., van der Linden, C. J., Leeuwenberg, J. F. M. & Buurman, W. A. A Comparison of Substrates for Human Umbilical Vein Endothelial Cell Culture. *Biotech. Histochem.* **67**, 241–250 (1992).
29. Akther, F., Yakob, S. B., Nguyen, N.-T. & Ta, H. T. Surface Modification Techniques for Endothelial Cell Seeding in PDMS Microfluidic Devices. *Biosensors* **10**, 182 (2020).
30. Relou, I. A. M., Damen, C. A., van der Schaft, D. W. J., Groenewegen, G. & Griffioen, A. W. Effect of culture conditions on endothelial cell growth and responsiveness. *Tissue Cell* **30**, 525–530 (1998).
31. Leclech, C. & Barakat, A. I. Is there a universal mechanism of cell alignment in response to substrate topography? *Cytoskeleton* (2021). doi:10.1002/cm.21661
32. Leclech, C. & Villard, C. Cellular and Subcellular Contact Guidance on Microfabricated Substrates. *Frontiers in Bioengineering and Biotechnology* 8–551505 (2020). doi:10.3389/fbioe.2020.551505
33. Natale, C. F., Lafaurie-Janvore, J., Ventre, M., Babataheri, A. & Barakat, A. I. Focal adhesion clustering drives endothelial cell morphology on patterned surfaces. *J. R. Soc. Interface* **16**, 20190263 (2019).
34. Duclos, G. *et al.* Spontaneous shear flow in confined cellular nematics. *Nat. Phys.* **14**, 728–732 (2018).
35. Marchetti, M. C. *et al.* Hydrodynamics of soft active matter. *Rev. Mod. Phys.* **85**, 1143–1189 (2013).
36. Thampi, S. P., Doostmohammadi, A., Golestanian, R. & Yeomans, J. M. Intrinsic free energy in active nematics. *EPL (Europhysics Lett.)* **112**, 28004 (2015).
37. Prost, J., Joanny, J.-F. & Parrondo, J. M. R. Generalized Fluctuation-Dissipation

- Theorem for Steady-State Systems. *Phys. Rev. Lett.* **103**, 090601 (2009).
38. Cengio, S. D., Levis, D. & Pagonabarraga, I. Fluctuation–dissipation relations in the absence of detailed balance: formalism and applications to active matter. *J. Stat. Mech. Theory Exp.* **2021**, 043201 (2021).
 39. Sharma, A. & Ruckenstein, E. An analytical nonlinear theory of thin film rupture and its application to wetting films. *J. Colloid Interface Sci.* **113**, 456–479 (1986).

REVIEWER COMMENTS

Reviewer #1 (Remarks to the Author):

The authors have replied to all the points raised in the previous round of revision, by including new experiments or discussing in the manuscript text.

While not all questions I posed have been fully clarified, I understand that some may require a disproportionate amount of work for performing in silico or in vitro experiments.

The absence of a clear parallelism or function in vivo for the interesting phenomenon described limits its relevance, which is however sufficiently new to appear in Nat Comm.

Reviewer #2 (Remarks to the Author):

The authors provide very detailed answers to the questions raised by the referees. They also revised strongly the theoretical interpretation of the experimental results and they now perform a non-linear stability analysis. Most of the answers to the points that I raised are convincing but I think that the presentation of the non-linear stability analysis must be clarified prior to publication.

If I understand well, the authors try to determine the steady state flow and orientation patterns above a bifurcation from a non-flowing state to a flowing state when the activity ζ is increased. They consider a weakly non-linear theory where the variations of the angle θ and of the velocity gradient are small. This weakly non linear theory can be expected to be accurate only close to the bifurcation threshold ie when the activity is close to the critical value ζ_c corresponding to value where the growth rate equation obtained when equation 9 vanishes start to have solutions at finite wave vectors. The numerical estimates given by the authors show that the system is far from the bifurcation threshold and that ζ is much larger than ζ_c .

Still one can proceed with the weakly non-linear calculation. Despite the fact that it will not be fully accurate, it will give at least the qualitative behavior. I am not sure that I understand well the procedure used by the author, which is adapted from reference 9.

An alternative way to calculate the steady state would be to use the procedure outlined in reference 6. If the friction vanishes for instance, the problem can be mapped onto the motion of a particle of position θ in a non-harmonic potential and the weakly non-linear theory follows the lines of the classical weakly non-linear solution of the pendulum equation. The current problem is more complicated due to the friction but can probably be solved using the same method. In the end the full calculation provides not only the change in the steady state wave vector with the amplitude of the angle δ (equation 22) but it also provides explicitly the value of the amplitude δ (or ϵ) as a function of the activity ζ . This does not seem to be the case here. The authors only say that ϵ should be proportional to the orientational field α .

Why did the author not calculate explicitly δ or ϵ and use the results of this explicit calculation to compare to the experiments.

Response to Reviewer 2

The authors provide very detailed answers to the questions raised by the referees. They also revised strongly the theoretical interpretation of the experimental results and they now perform a non-linear stability analysis. Most of the answers to the points that I raised are convincing but I think that the presentation of the non-linear stability analysis must be clarified prior to publication.

If I understand well, the authors try to determine the steady state flow and orientation patterns above a bifurcation from a non-flowing state to a flowing state when the activity ζ is increased. They consider a weakly non-linear theory where the variations of the angle θ and of the velocity gradient are small. This weakly non linear theory can be expected to be accurate only close to the bifurcation threshold ie when the activity is close to the critical value ζ_c corresponding to value where the growth rate equation obtained when equation 9 vanishes start to have solutions at finite wave vectors.

The numerical estimates given by the authors show that the system is far from the bifurcation threshold and that ζ is much larger than ζ_c .

Still one can proceed with the weakly non-linear calculation. Despite the fact that it will not be fully accurate, it will give at least the qualitative behavior. I am not sure that I understand well the procedure used by the author, which is adapted from reference 9.

An alternative way to calculate the steady state would be to use the procedure outlined in reference 6. If the friction vanishes for instance, the problem can be mapped onto the motion of a particle of position θ in a non-harmonic potential and the weakly non-linear theory follows the lines of the classical weakly non-linear solution of the pendulum equation. The current problem is more complicated due to the friction but can probably be solved using the same method. In the end the full calculation provides not only the change in the steady state wave vector with the amplitude of the angle δ (equation 22) but it also provides explicitly the value of the amplitude δ (or ϵ) as a function of the activity ζ . This does not seem to be the case here. The authors only say that ϵ should be proportional to the orientational field α .

Why did the author not calculate explicitly δ or ϵ and use the results of this explicit calculation to compare to the experiments.

We thank the Reviewer for his/her appreciation of the revised version as well as for the additional comment, which has led us to further improve the analysis.

As the Reviewer indicates, our experimental conditions correspond to a level of contractility (activity) that is much larger than the bifurcation threshold. Thus, unlike previous studies, we do not seek to characterize the system close to the bifurcation threshold. Rather, our experiment starts from an unstable equilibrium state at a level of cell activity significantly above the threshold, and we investigate how the unstable perturbations grow over time. We use a nonlinear stability analysis to get an indication of how the wavelength of the dominant perturbation evolves as perturbations grow in amplitude, and the result agrees with experiments. The “small parameter” in our perturbation analysis is the amplitude of the perturbation itself, not $(\zeta - \zeta_c)$.

We also emphasize that previous studies mentioned by the Reviewer (Refs. 5 and 6) investigated systems of finite width W . In such case, there is a bifurcation threshold at a certain critical width W_c . These earlier studies considered systems where W was slightly larger than W_c , a regime that is readily accessible experimentally. In contrast, we investigate systems of virtually infinite width. The only bifurcation threshold that is relevant to us corresponds to the case where cell activity (contractility) ζ is slightly larger than ζ_c . However, unlike the parameter W , measuring ζ , or varying it in a controlled manner, is experimentally challenging, and a comparison to experiments is no longer straightforward.

In spite of these caveats, we have performed the additional analysis suggested by the Reviewer, which is presented in the revised Supplementary Note. The analysis allows us to estimate the contractility (activity) corresponding to the bifurcation threshold. As anticipated, this critical contractility is about one order of magnitude smaller than actual cell contractility in our experiments. For this reason, the results of this new analysis are inconsistent with our experiments.

In conclusion, we believe that the non-linear analysis used in the manuscript is more suited for our experimental conditions than the proposed bifurcation analysis. As the bifurcation analysis can nevertheless be interesting to the readers and supports the notion that cell contractility in our experiments is far above critical, we have included it in the revised Supplementary Note. To emphasize this point, which justifies the application of the nonlinear analysis method in Ref. 9, we have additionally edited the Supplementary Note to more explicitly state that our stability analysis describes the growth of perturbations of an initial base state that is far from the bifurcation threshold, as is the case in our experiments. The method we used is described in detail in Ref. 9, which we indicate in the revised Supplementary Note. As this constitutes a rather technical discussion for specialized readers, we believe it is not mandatory to include it in the main text, as it would unnecessarily muddle the main message.

REVIEWERS' COMMENTS

Reviewer #2 (Remarks to the Author):

The responses of the authors to my previous comments are convincing and I recommend the publication of this revised version. I do agree with the authors that the new technical discussion should remain in the supplementary material only.